# Assessment of Smell and Taste Disturbances among COVID-19 Convalescent Patients: A Cross-Sectional Study in Armenia

**DOI:** 10.3390/jcm11123313

**Published:** 2022-06-09

**Authors:** Karine Melkumyan, Darshan Shingala, Syuzanna Simonyan, Hrag Torossian, Karen Mkrtumyan, Karen Dilbaryan, Garri Davtyan, Erik Vardumyan, Konstantin Yenkoyan

**Affiliations:** 1Neuroscience Laboratory, COBRAIN Center, Yerevan State Medical University after Mkhitar Heratsi, Yerevan 0025, Armenia; karine.melkumyan@gmail.com (K.M.); hntorossian@gmail.com (H.T.); karendilb@gmail.com (K.D.); 2Department of Physiology, Yerevan State Medical University after Mkhitar Heratsi, Yerevan 0025, Armenia; 3General Medicine Faculty, Yerevan State Medical University after Mkhitar Heratsi, Yerevan 0025, Armenia; shingala.darshan@gmail.com (D.S.); syuzi.simonyan1998@mail.ru (S.S.); gardavits@gmail.com (G.D.); erik.vardum@gmail.com (E.V.); 4ClinChoice LLC., Yerevan 0033, Armenia; 5Krisp Technologies Inc., Yerevan 0033, Armenia; kmkrtumyan@krisp.ai; 6Department of Pharmacology, Yerevan State Medical University after Mkhitar Heratsi, Yerevan 0025, Armenia; 7Department of Bochemistry, Yerevan State Medical University after Mkhitar Heratsi, Yerevan 0025, Armenia

**Keywords:** COVID-19, smell disturbances, taste disturbances, headache, olfactory nerve, trigeminal nerve, neurosensory dysfunction, diagnostic marker

## Abstract

Background and Objectives: Neurological manifestations of Coronavirus Disease 2019 (COVID-19) such as olfactory and gustatory disturbance have been reported among convalescent COVID-19 patients. However, scientific data on the prevalence of smell and taste disturbance are lacking. Therefore, we present findings on the degree of smell and taste disturbances among the Armenian population. Methods: Study participants were randomly recruited and then categorized into two groups based on their course of the disease. A cross-sectional study was performed to assess participants’ sensitivity to smell triggered by the olfactory and the trigeminal nerves; their ability to differentiate between various odors; and to evaluate their gustatory perception. Results: The smell test revealed that the degree of olfactory nerve disturbance was different by 30.7% in those participants of the early group as compared to those of the late group, and the degree of trigeminal nerve disturbance was different by 71.3% in the early group as compared to the late group. A variation of the differentiating ability among the participants of the early and late groups was detected. Gustatory disturbances for all flavors were also found to be different in both the groups. A moderate positive correlation (0.51) was found between the overall sensitivity of smell and the ability to differentiate between various odors as cumulatively stimulated by both the olfactory and trigeminal nerves. Also, a moderate positive correlation (0.33) was found between headache and smell sensitivity through the olfactory nerve and a high negative correlation (−0.71) was found between headache and smell sensitivity through the trigeminal nerve. Conclusion: Pathological changes in the olfactory and trigeminal perceptive abilities caused disturbances in smell sensation, with the trigeminal nerve being more affected. The capacity to differentiate fragrances did not improve with time and the disturbance severity of bitter taste perception was higher among the study participants.

## 1. Introduction

Coronavirus Disease 2019 (COVID-19) has been associated with a variety of neurological manifestations ranging from loss of smell and taste sensation, weakened concentration, arthralgia, myalgia, impaired circadian rhythm, headache, encephalitis, and stroke to psychological effects such as depression or psychosis [1,2]. These neurological signs and symptoms due to COVID-19 infection can be further classified based on the severity of the disease and/or course of infection. Symptoms such as hyposmia/anosmia and hypogeusia/ageusia are commonly found in mild cases, whereas symptoms such as stroke, delirium, and neuronal inflammation are usually reported in severe cases [3,4,5,6].

Neurosensory dysfunction such as loss of smell (hyposmia) and taste (hypogeusia) caused by SARS-CoV-2 are generally less harmful than pulmonary symptoms [7]. Anecdotal evidence from researchers and healthcare providers worldwide proposes hyposmia and hypogeusia as a hallmark for suspecting COVID-19 because the transient abnormal change in taste and smell sensation occurs before other systematic manifestations [8,9]. Therefore, the onset of such neurosensory dysfunction can be helpful for early diagnosis, effective management and better prognosis of patients suffering from COVID-19 [10].

The pathogenies of Severe Acute Respiratory Syndrome Coronavirus 2 (SARS-CoV-2) infecting brain cells remain elusive; however, the neurological manifestations are presumed to be instigated because of a local inflammatory response, focal immune response to neuroinflammation, and impairment of neural vasculature [11]. In addition to that, speculations such as adverse effects due to certain drug therapies, involvement of the central nervous system (CNS), affection of the peripheral nervous system (PNS) with specific influence on cranial nerves I, VII, IX, and X have been documented to explain the neuro-invasion of SARS-CoV-2 [12,13,14,15,16].

Existing scientific evidence also suggests that SARS-CoV-2 could possibly bind to the angiotensin-converting-enzyme-2 (ACE2) in the nasal mucosal membrane [17]. However, it is important to note that neurosensory dysfunction in COVID-19 infected patients may occur without signs of nasal obstruction, hence, Jafari et al., proposed that SARS-CoV-2 can likely invade olfactory and gustatory neurons and cause neurotoxicity leading to hyposmia and hypogeusia [18].

According to Matschke et. al, SARS-CoV-2 protein and ribonucleic acid (RNA) have been detected in neural tissues and cerebrospinal fluid (CSF) of COVID infected patients [14,19]. Scientific studies indicate that SARS-CoV-2 has the potential to invade the human nervous system [20]. In a systematic review of 24 studies conducted by Agyeman et al., 41% and 38% of COVID-19 infected patients were reported to have had either olfactory or gustatory perceptive disorders, respectively [14]. Although it is well established that SARS-CoV-2 does not exclusively affect the pulmonary system [21], COVID-19-induced anosmia is an underestimated symptom and may help to understand the underlying pathological mechanism of the virus [15].

The uncertainty of pathophysiological mechanisms and significant variability in neurosensory dysfunction, such as a loss of smell (hyposmia) and taste (hypogeusia) caused by SARS-CoV-2 have been associated with its neuro-invasion [6,22]. However, there is a scarcity of scientific data related to the prevalence of these symptoms in COVID-19 convalescent patients. The aim of this cross-sectional study was to acquire an insight into the pandemic si tuation in Armenia, and to assess the status of taste and smell disturbances among COVID-19 convalescent patients. Our objectives were: to estimate the prevalence and characteristics of gustatory dysfunction; to identify differences in smell sensation triggered by olfactory and trigeminal nerves; to find an association between headache and sensitivity of smell sensation through olfactory and trigeminal nerves; to find a correlation between sensitivity and differentiation of smell sensation; and to evaluate the differences in taste and smell disturbances among COVID-19 infected patients before and after 130 days of disease diagnosis.

## 2. Methodology

### 2.1. Study Design

We conducted a cross-sectional study, which had three main assessment criteria, namely, assessment of smell sensation through sensitivity tests of olfactory and trigeminal nerves, assessment of the ability to differentiate various odors, and assessment of the level of taste perception and differentiation. In addition to that, the participants’ COVID severity index was assessed based on the symptoms of pneumonia and oxygen saturation (SpO2) during their initial weeks of diagnosis [3,23,24,25].

The research methodology and study instruments were based on the University of Pennsylvania Smell Identification Test (UPSIT) and snip-and-sniff test [20]. However, since the research methodology and study instruments were modified and adapted to be most appropriate for the Armenian context, the researchers designed a pretest at the Neuro-science Laboratory (COBRAIN Center, YSMU) and executed the pretest among 100 practically healthy volunteers who had not been infected with SARS-CoV-2 at the first hospital complex of the Yerevan State Medical University after Mkhitar Heratsi. The results derived from the pretest were used to define criteria for smell and taste sensitivity disturbances; categorize other study variables; and set threshold levels. 

#### 2.1.1. Assessment of Smell Sensation through Sensitivity Tests of Olfactory and Trigeminal Nerves

Smell sensation is mainly perceived through two cranial nerves, that is, the olfactory and the trigeminal nerves [26,27,28]. It has been found that a majority of fragrances are recognized by both the nerves; however, there are a few orders specifically perceived by the olfactory or the trigeminal nerve [29]. For instance, the olfactory receptor neuron (mOR-EG), has been found to be triggered by eugenol containing essential oils such as clove oils [3,24,30] and the trigeminal receptor neuron (TRPV3), has been found to identify the odor of camphor spirit as temperature [24,25,26,27,28,29,30,31,32,33]. In addition to that, eugenol is also found to be a pure olfactory or a non-trigeminal stimulus [34]. Therefore, we decided to use clove essential oil to determine the sensitivity levels of the olfactory nerve and camphor spirit to determine the sensitivity of the trigeminal nerve.

To investigate the sensitivity levels of the olfactory and trigeminal nerve, standard 20 mL pet opaque bottle containers were used. Overall, eighteen such containers were used in a set; from which eight containers were filled with different concentrations of clove essential oil to examine the sensitivity levels of the olfactory nerve; another eight containers were filled with different concentrations of camphor spirit to examine the sensitivity of the trigeminal nerve; the remaining two bottles were filled with distilled water as odorless variants to ensure smell sensation. 

To determine the sensitivity levels of the olfactory nerve, the first bottle contained pure clove essential oil with the highest concentration of odor. This was diluted by a ratio of 1:10 to fill the consequent seven bottles. The dilution was achieved by adding distilled water. Hence, the second bottle was filled with 1ml oil from the content of the first bottle and diluted with 10ml of distilled water; this was repeated consecutively until the eighth concentration was achieved. Similarly, to determine the sensitivity of the trigeminal nerve, an equivalent concentration system was used. For the trigeminal nerve, 10% of camphor spirit was used, and the remaining containers were diluted by a 1:10 ratio using distilled water. 

In both sets, the eighth dilution was prepared in such a manner that 75% of healthy individuals can perceive the smells, and the content of the seventh bottle can easily be sensed by 100% of the healthy population, based on results from pretest participants. These thresholds were pretested in 100 volunteers who had not been infected with SARS-CoV-2 and were considered to be practically healthy.

The participants were requested to close their eyes, and a dilution sequence of three bottles was held under their nose in a random order. One of the bottles was the one with the odor; the other two were the odorless variants. Each bottle was held 2 mm under the nostrils without touching them for 4–5 s, and the investigator asked the participant to smell the contents. Each set dilution sequence was presented to the participant after a pause of 45 s to avoid fragrance remanence from the previous concentration. 

After smelling a set of three bottles, the participant had to identify which was the odor contained in the bottle, based on their sense of smell. With this principle, each set of dilution sequences was introduced thrice and if the participant guessed the correct bottle at least twice, that concentration assessment was considered positive and the investigator continued to the next dilution sequence. This was continued until reaching the dilution for which the participant gave two negative guesses. In each test, the investigator started with the median (4th container) concentration and moved accordingly; if the participant was considered positive for this, the investigator moved to the lesser concentration (higher number); if negative, the investigator moved to the higher concentration (lesser number). The sensitivity level of the olfactory and trigeminal nerves was assessed with a 0–8 scoring system each. In case the participant did not sense any of the given odors, they were given a score of 0 and depending on which concentration they sensed successfully, they were given a score ranging between 1 and 8. 

#### 2.1.2. Assessment of the Ability to Differentiate Various Odors

To access the ability to distinguish different odors, sixteen fragrances were selected according to the customs of the Armenian population and their familiarity with fragrances such as: acetone, alcohol, basil, chocolate, cigarette, cinnamon, clove, coffee, garlic, menthol, orange, rose, thyme, valerian, vanilla, and vinegar. All sixteen odors were prepared from standardized high-quality cold-pressed essential oils and fresh organic derivatives; for instance, Vivasan Swiss-made essential oils manufactured in Sägehüslistrasse 10, 9050 Appenzell, Switzerland for clove, rose, menthol, thyme, basil, and orange; Biofinest USA-made garlic essential oil manufactured in Lindon United States was used and Nature-hue Chinese-made cinnamon essential oil manufactured in Newark, United States were used. Lab-quality alcohol (96%) and acetone were used to assess the smell of alcohol and acetone; organic apple cider vinegar (5%), vanilla and valerian were used to assess the smell of vinegar, vanilla and valerian; freshly ground Arabica coffee beans derivative was used to assess the coffee smell; cocoa powder and cigarette butts were used to assess the smell of chocolate and cigarette. 

All these odors were numbered sequentially from one to sixteen and were placed in uniform containers. For investigation, each participant was requested to shut their eyes during the assessment. Each container was held 2 mm under the nostrils for about 4–5 s for the participant to smell the contents. Participants were asked to identify the fragrance; however, if needed, they were provided with six options to select from, namely, the correct answer, three different odors, no odor at all, and smell without any differentiation. Each odorant container was presented to the participant after a pause of 45 s to avoid fragrance remanence from the previous container. A binary score scale was used as an indicator of the ability to differentiate various odors. 

The methodology of sensitivity tests conducted to evaluate the smell sensations of olfactory and trigeminal nerves were adapted from the University of Pennsylvania Smell Identification Test (UPSIT) and a snip-and-sniff test [35]. The methodology was modified to fit the Armenian context during the pandemic and was adapted at the Neuroscience Laboratory (COBRAIN Center, YSMU). 

#### 2.1.3. Assessment of the Level of Taste Perception and Differentiation

The assessment of taste perception and differentiation was determined using four primary flavors, that is, sweet, salty, sour, and bitter. A total of twelve containers, three per flavor, were used to evaluate the taste test. Three different solutions with varying concentrations were used for each taste; dilute, intermediate, and concentrated. All the solutions were fed in order of dilution and flavor profile starting from sweet, to salty, then sour, and then bitter, to avoid aftertaste and prevent flavor interference [36,37].

For investigation, participants were requested to open their mouths so that the investigator can place a few drops of flavored solution using a disposable pipette.

In case the participant identified the dilute solution, 3 points were recorded and the investigator continued to the next taste. In case the participant failed to identify the flavor, the participant was requested to rinse their mouth and/or drink water before trying the intermediate concentration of the same flavor. If the participant identified the flavor correctly, a score of 2 points was recorded and in case of an incorrect answer, the investigator continued the test with the concentrated solution and recorded 1 point if correctly recognized. In case the participant failed to identify flavor in any of the three concentrations, 0 points was recorded. The scoring criteria was based on previously established literature [22].

All the solutions for the smell and taste tests were freshly prepared in a sterile environment before starting the investigation process. The orifice of glass containers was sterilized with 96% ethanol solution after investigating each participant and disposable pipettes were used for each participant. Table 1 presents the solutions used for taste assessment.

### 2.2. Study Population 

The target population was native adult Armenians who self-reported subjective disturbances in the perception of smell and/or taste sensations 14 days following a COVID-19 diagnosis as confirmed by a positive PCR test at the time of diagnosis. Taking into consideration the effects of ageing on olfactory and gustatory function, the age limit for inclusion in the study was determined to be from 18 to 65 years inclusive [38]. Results of anti-SARS-CoV-2 antibodies were expressed either as negative (cut-off index < 1) or positive (cut-off index ≥ 1) with the cut-off indices being determined as per the most recent literature available at the time of the study [39]. Participants with negative anti-SARS-CoV-2 antibodies were excluded from the study. Participants with a self-reported known history of comorbidities such as active allergies, acute rhinitis, neurodegenerative disorders etc. and presence of smell and/or taste disturbances prior to COVID-19 diagnosis due to other known causes such as recent rhinoplasty, trauma etc. were excluded from the study.

The following inclusion/exclusion criteria were used to determine the eligibility of study participants:

Inclusion criteria:Age ≥ 18 years and ≤65 yearsArmenian nationalitySubjective presence of smell and/or taste disturbances upon presentationPositive SARS-CoV-2 PCR test at the time of COVID-19 diagnosis

Exclusion criteria:Anti-SARS-CoV-2 antibodies < 1 cutoff indexSmell and/or taste disturbances present before COVID-19 diagnosis due to other causes such as recent rhinoplasty, traumas etc.Presence of comorbidities such as active allergies, acute rhinitis, neurodegenerative disorders etc.

A total of 223 participants signed-up using a “Google form” shared via social media platforms to approach a large audience. After successfully signing up, all study participants were contacted by the recruitment team using the provided phone numbers and were evaluated against the above-mentioned eligibility criteria. Of 223 participants, 7 were not included due to a failure to produce a positive SARS-CoV-2 PCR test report at the time of COVID-19 diagnosis, and 2 were not included as they were above 65 years of age. Based on the exclusion criteria, 10 participants were excluded since their anti-SARS-CoV-2 antibodies cutoff was below 1; 1 was excluded due to the presence of smell and/or taste disturbances following recent rhinoplasty prior to COVID-19 diagnosis, and 1 was excluded due to the presence of a comorbid neurodegenerative disorder. As a result, a total of 202 participants were successfully enrolled in the study.

All study participants (*n* = 202), were further divided into two groups, namely “Early” and “Late” based on the time interval between their positive COVID-19 diagnosis and presentation to the first hospital complex of the Yerevan State Medical University after Mkhitar Heratsi for study investigation. Participants who were investigated before the 130th day since their positive COVID-19 diagnosis were included in the “Early” group and participants who were investigated after the 130th day since their positive COVID-19 diagnosis were included in the “Late” group [40]. Figure 1 represents the study participants flow chart graphically.

### 2.3. Sample Size Calculation 

We calculated the sample size using the level of significance of 0.05 (*α*), power of 80% and the following formula comparing proportions in two independent equal groups:$$n={\left(z_{\alpha /2}+z_{\beta }\right)}^{2}\frac{\;\left\{p_{1}\left(1-p_{1}\right)+p_{2}\left(1-p_{2}\;\right)\right\}\;}{{\left(p_{1}-p_{2}\right)}^{2}}$$

Taking into account that the majority of literature-based evidences is derived from scientific studies conducted in high-income countries, therefore, calculating a reliable proportion of the estimated amount of required participants for a lower-middle-income nation such as Armenia was quite challenging [15]. Hence, by using a conservative technique, we assumed that *p*_1_ = 60% and *p*_2_ = 40% which was inputted into the above-mentioned formula to obtain *n* = (1.96 + 0.84)^2^ [{0.61 − 0.6 + 0.41 − 0.4}/(0.2)^2^] = 95 per group, implicating a total of minimum *n* = 190.

### 2.4. Sampling Strategy

A sign-up form using “Google forms” was prepared and shared on social media platforms to approach a large audience. After successfully signing up, all study participants were contacted by the recruitment team using their phone numbers and checked for their eligibility. In the case of successfully corresponding to the eligibility criteria, the participant was invited to the first hospital complex of the Yerevan State Medical University after Mkhitar Heratsi where the investigations were taking place, and an informed written consent was obtained from each participant. Each participant was incentivized with a handmade aromatic reed diffuser to appreciate their time and participation in the study. 

### 2.5. Study Instrument

An interviewer-administered paper-based study instrument was used to collect data. It was developed, modified, and adapted as per the Armenian context from a pre-existing methodology. The maximum possible score a participant could obtain was 44 points. The study instrument consisted of four main components, which were as follows:Demographic and symptomatic information of the participant

Standard demographic information including age, sex, height, and weight along with symptomatic information such as fever, malaise, fatigue, headache etc. was obtained from the study participants. 

2.A sensitivity test to assess the sensation of smell as triggered by olfactory and trigeminal nerves

The participants were given a score of 0 for each incorrect answer and their response was categorized as anosmia. For every correct answer, the participants were scored from 1 to 8 in the sensitivity test for each corresponding nerve and their response was categorized as normosmia for a score of 7–8, mild hyposmia for a score of 5–6, moderate hyposmia for a score of 3–4, severe hyposmia for a score of 1–2, and anosmia for a score of 0. The findings from the pretest revealed that approximately 3% of the pretest participants constituting practically healthy individuals who had not been infected with SARS-CoV-2 were found to have smell disturbances. Based on this finding and taking the margin of error into consideration, we categorized the level of smell disturbances as normosmia, mild hyposmia, moderate hyposmia, severe hyposmia, and anosmia, and defined the threshold ranges and cut-off indices as 7–8, 5–6, 3–4, 1–2 and 0 respectively [41]. Our pretest results are presented in Appendix A.

3.Differentiation tests to assess the ability to differentiate various odors.

The participants were given a score of 0 for each incorrect answer and a score of 1 for each correct answer. The responses were recorded as a continuous variable ranging from 0 to 16.

4.Taste test to assess the level of taste perception and differentiation.

The participants were given a score of 0 for each incorrect answer and their response was categorized as severe hypogeusia and/or ageusia. For every correct answer, the participants were scored from 1 to 3 for each corresponding taste, that is, sweet, salty, sour, or bitter. The responses for each taste were categorized as normogeusia for a score of 3, mild hypogeusia for a score of 2, and moderate hypogeusia for a score of 1. These were based on previously established criteria [42].

### 2.6. Study Variables

The primary outcome variable was dichotomous, that is, COVID-19 convalescent patients who participated within 130 days of diagnosis versus COVID-19 convalescent patients who participated after 130 days of diagnosis.

Symptoms of COVID (binary), olfactory nerve sensitivity (categorical), trigeminal nerve sensitivity (categorical), smell differentiation (continuous), and taste perception (categorical).

### 2.7. Statistical Analysis

IBM SPSS software 26 (Yerevan State Medical University after Mkhitar Heratsi, Yerevan, Armenia) was used for data entry and data cleaning; STATA 16.0 and Python 3.9.7 (Yerevan State Medical University after Mkhitar Heratsi, Yerevan, Armenia) were used for conducting statistical analyses.

The normally distributed demographic variables were presented with means and standard deviations (SDs). The associations between the olfactory, and gustatory outcomes were analyzed using cross-tab and chi-square tests between two categorical variables organized in a bivariate table [43]. The analyses of smell sensitivity (categorical) and smell differentiation (categorical) of olfactory and trigeminal nerves was performed using chi-square test and logistic regression with a level of significance of α = 0.05. Correlation between the presence or absence of headaches (binary) and smell sensitivity/differentiation capacity (continuous) was investigated using a point-biserial Pearson correlation coefficient with a 95% confidence interval (CI). A heatmap correlation matrix was also conducted between all variables of the study, that is, smell sensitivity and differentiation; taste sensitivity and differentiation; and prevalence of headaches.

## 3. Ethical Considerations

The study received an approval from the Institutional Review Board (IRB) of Yerevan State Medical University named after Mkhitar Heratsi (Identification code N 8-2/20. Date: 2 June 2020). An informed consent confirmed by the YSMU Ethical Committee was obtained from the participants prior to the investigation. The consent form had several components including the right to privacy, and the right to refuse participation at any point in time without any losses or benefits. Each participant was assigned a specific patient ID at the end of the assessment to maintain their confidentiality. Only the investigators were granted access to the tools, equipment, documents, data and papers pertaining to the study. Personal identifiable information such as the names and phone numbers of participants were kept confidential in an encrypted folder which was only accessible to the principal investigator.

## 4. Results

### 4.1. Descriptive Analysis

Overall, 202 participants were eligible to take part in the study based on the inclusion/exclusion criteria. The mean age of the study participants was 37.04 ± 11.82 years ranging from 18 years to 65 years. A total of 151 study participants were females accounting for 74.75%. About 64.25% of participants reported during the investigation that they had a headache and the majority of participants (90%) participants were diagnosed with a mild form of pneumonia when they were diagnosed and infected with COVID-19. As per the participants’ COVID severity index assessed based on the symptoms of pneumonia and oxygen saturation (SpO2) during their initial weeks of diagnosis, the majority of participants (90.1%) had a mild form of infection, 5.4% had moderate symptoms and 4.5% had a severe infection. None of the participants were hospitalized, had to stay in the critical unit (ICU) for observation, or had pulmonary complications post-COVID-19 infection. Table 2 represents the demographic data in detail. 

### 4.2. Smell Sensitivity and Smell Differentiation Tests of Olfactory and Trigeminal Nerves 

The sensory tests of the olfactory nerve revealed that 69.31% of the participants had normosmia, 16.83% had mild hyposmia, 5.94% had moderate hyposmia, 5.45% had severe hyposmia, and 2.48% had anosmia. The sensory tests of the trigeminal nerve revealed that 28.71% of the participants had normosmia, 13.37% had mild hyposmia, 27.72% had moderate hyposmia, 27.23% had severe hyposmia, and 2.97% had anosmia. 

For statistical analysis of the data, chi-square test and logistic regression were used at a level of significance of α = 0.05. The analysis suggests that the degree of disturbance in both the nerves, that is, trigeminal and olfactory, was significantly different (*p*-value < 0.000) as represented in Figure 2. 

The study participants (*n* = 202) were divided into two groups based on the date of diagnosis and their visit for investigation. Participants who came for the investigation within 130 days since their positive COVID-19 diagnosis were classified as the early group (n_1_ = 102; 50.50%) and the participants who came for the investigation after 130 days since their positive COVID-19 diagnosis were classified as the late group (n_2_ = 100; 49.50%). 

The sensory tests of the olfactory nerve revealed that 59% of the participants had normosmia, 23% had mild hyposmia, 8% had moderate hyposmia, 6% had severe hyposmia, and 4% had anosmia in the early group, and 79.41% of the participants had normosmia, 10.78% had mild hyposmia, 3.92% had moderate hyposmia, 4.90% had severe hyposmia, and 0.98% had anosmia among the late group. The level of normosmia for the olfactory nerve increased by 20.41 units in the late group as compared to the early group as represented in Figure 3A. The analysis suggests that the degree of disturbance in the olfactory nerve was significantly different (*p*-value = 0.028) in those participants who came within 130 days of diagnosis as compared to those who came after 130 days. 

The sensory tests of the trigeminal nerve revealed that 22% of the participants had normosmia, 14% had mild hyposmia, 27% had moderate hyposmia, 33% had severe hyposmia, and 4% had anosmia in the early group, and 35.29% of the participants had normosmia, 12.75% had mild hyposmia, 28.43% had moderate hyposmia, 21.57% had severe hyposmia, and 1.96% had anosmia in the late group. The level of normosmia for the trigeminal nerve increased by 13.29 units in the late group as compared to the early group as represented in Figure 3B. The analysis suggests that the degree of disturbance in the trigeminal nerve was not significantly different (*p*-value = 0.175) in those participants who came within 130 days of diagnosis as compared to those who came after 130 days. 

A moderate correlation of 0.51 was found between the overall sensitivity of smell and ability to differentiate between various odors as cumulatively stimulated by both the olfactory and trigeminal nerves (*p*-value < 0.000) as represented in Figure 4. 

As represented in Figure 5, we evaluated the differences in the overall sensitivity of smell and ability to differentiate between various odors as cumulatively stimulated by both olfactory and trigeminal nerves among participants of the early and late groups. The cumulative mean of the sensitivity of smell stimulated by both the olfactory and trigeminal nerves was 10.19 in the early group and 11.96 in the late group. The cumulative mean of the ability to differentiate between various odors stimulated by both the olfactory and trigeminal nerves was 10.43 in the early group and 9.33 in the late group. 

We could also detect a variation of the differentiating ability among the participants of the early and late groups. This is presented in Figure 6.

A moderate negative correlation of 0.33 was found between the headache and sensitivity of smell perception through olfactory nerves and a moderately high negative correlation of 0.71 was found between the headache and sensitivity of smell perception through trigeminal nerves of the participants (*p*-value < 0.000). Hence, the correlation of headache with the sensitivity of smell perception through the trigeminal nerve was two times higher.

### 4.3. Gustatory Tests for Sweet, Salty, Sour, and Bitter Tastes

The gustatory test for sweet taste revealed that 50% of the participants had normogeusia, 44.55% had mild hypogeusia, 3.96% had moderate hypogeusia, and 1.49% had severe hypogeusia and/or ageusia. The gustatory test for salty taste revealed that 75.25% of the participants had normogeusia, 15.84% had mild hypogeusia, 5.94% had moderate hypogeusia, and 2.97% had severe hypogeusia and/or ageusia. The gustatory test for sour taste revealed that 79.7% of the participants had normogeusia, 16.34% had mild hypogeusia, 2.97% had moderate hypogeusia, and 0.99% had severe hypogeusia and/or ageusia. The gustatory test for bitter taste revealed that 54.95% of the participants had normogeusia, 21.29% had mild hypogeusia, 10.40% had moderate hypogeusia, and 13.97% had severe hypogeusia and/or ageusia. Figure 7 represents the gustatory data in detail. Overlapping of smell and taste disturbances among the early and late groups was found with 75 (73.5%) having both in early and 66 (66%) in late.

A Pearson point-biserial correlation analysis was performed between smell sensitivity, differentiation capacity, and the prevalence of headaches among the participants of the study. For olfactory nerve and headache, Point-Biserial (r) = 0.3260146 with Confidence Interval (95% CI: 0.197814–0.437907). For the trigeminal nerve and headache, Point-Biserial (r) = 0.7107635 with Confidence Interval (95% CI: 0.6428661–0.7628575). A heatmap correlation matrix was conducted between smell sensitivity and differentiation; taste sensitivity and differentiation; and prevalence of headaches. The correlation matrix is graphically represented in Figure 8.

## 5. Discussion

Since our study participants came for investigation after disease diagnosis at different time intervals, we expected their quantitative and qualitative characteristics would be different because of the time interval. We decided to conduct a study to evaluate neurosensory disturbances with respect to time because several studies observed that, if COVID-associated olfactory and gustatory disturbances were not improved within the first two weeks of recovery, they will persist for a long period [26,44,45]. Also, a scientific experiment found that the anatomical and functional recovery of the olfactory epithelium takes about 45 days [46] and odor perception takes about 90 days to recover [47]. Considering this experiment, and the unclear mechanism of COVID-19, we further included 30 days for spontaneous recovery after COVID-19. Based on this assumption, we calculated an approximate number of days (~130) required for the olfactory epithelium to begin its recovery process post-COVID-19 infection. Hence, we classified our participants into two groups, i.e., the early group (participants who came for investigation within 130 days of COVID diagnosis) and late group (participants who came for investigation after 130 days of COVID diagnosis).

It is well known that the first cranial nerve is the focal nerve responsible for olfaction. It starts from the olfactory receptors extending from the olfactory foramina of the cribriform process. However, numerous nerve endings of the fifth cranial nerve, the trigeminal nerve, are also present in the nasal cavity. Since various studies have identified the presence of a huge viral load in the trigeminal ganglion [27,48,49] and its role in the perception and differentiation of certain odors [27,28], we decided to test this hypothesis of trigeminal nerve involvement by examining the sensitivity of the olfactory and the trigeminal nerve using clove and camphor fragrances. The rationale for choosing these specific fragrances was also based on various scientific studies, which suggested that the fragrances of cloves are primarily recognized by the olfactory nerve and the fragrances of camphor are primarily recognized by the trigeminal nerve [3,30,32,33].

After an assessment of smell sensation through sensitivity tests of the olfactory and the trigeminal nerves, we found that participants with disturbed smell sensation triggered by the trigeminal nerve were more severely disturbed than by the smell sensation triggered by the olfactory nerve. Our results were in line with the existing literature that the trigeminal nerve branches responsible for olfaction are affected more as compared to the olfactory nerve. This can be explained by the fact that the trigeminal nerve is located as free nerve endings and is more vulnerable to viral penetration compared with the olfactory nerve which starts as receptor. We also found a direct correlation between the sensitivity of the nerves and the differentiation of odors. 

Our observations of the early and late groups of the participants were interesting. For instance, we expected and discovered through our analyses that the sensitivity of the nerves is weaker in the early group participants as compared to the late group participants. However, we did not anticipate that the ability to differentiate scents would be weaker among the participants of the late group. 

Hence, we speculate that the sensitivity to smell recovered faster as compared to the ability to differentiate various fragrances. This also suggests that the central mechanisms are responsible for the pathogenesis of olfactory pathways, especially to recall and recognize different odors. Also, we found that the recognition of smells does not decrease proportionately for all fragrances because many study participants found it difficult to recognize the smell of chocolate. However, more scientific studies in this direction would provide more insight into this topic because several other studies found the mechanisms of olfactory disturbances to be either central or peripheral [6,16,25,50].

We also investigated the variation in taste perception for all four main flavors based on evidence suggesting that the taste disturbances can be flavor-dependent [6,51]. Our study results also indicate that the COVID-19-associated decline in taste sensation and perception is not uniform in all flavor profiles. We additionally found that the perception of bitter taste is most affected and the perception of sweet taste is least affected after COVID-19 infection. We also observed that the smell sensation was affected more than the taste perception because of COVID-19 infection, this result was also in line with the existing literature. We would also like to mention that as a majority of participants had mild COVID-19 infection, it would be appropriate to say that the results obtained are most fitting for the mild cases. To verify this, we compared presented results with data excluding moderate to severe cases and we had similar results. Also, the results are adjusted for variables such as severity of disease, age, gender and BMI. 

This study had several limitations, for instance, the use of in-house tests, non-validated study instruments, and modification of methodology are major drawbacks of this study. We have tried to minimize this by pretesting the questionnaire in a healthy population and preparing test materials in a sterile laboratory condition using standardized resources. Also, using a Google form to enroll participants potentially led to selection bias by limiting the reach to a larger population, especially those who belong to lower socio-economic status, and/or do not have access to the internet or mobile devices. In addition to that, subjective opinions for a few variables, such as the presence or absence of comorbidity and other neurological disturbances, have been noted; however, this data was not considered during data analysis. In addition to that, some unavoidable fragrances, such as environmental or room-borne odors could interfere with the perception of smell. We have tried to minimize this by requesting investigators and participants not to wear perfume to prevent the interference of fragrances. In addition to those, all the smell tests were conducted in a well-ventilated room, near an open window.

We recommend that a prospective study must be conducted on a large scale among COVID-19 convalescent patients using a single validated measure of smell and taste disturbances to provide a more accurate estimate of the intensity of olfactory and gustatory disturbances, taking into account all potential confounding factors. Evidence-based strategies should be recommended to people suffering from olfactory or gustatory disturbances following COVID-19 infection. Scientific literature suggests that olfactory training (OT) with high-concentration odors of phenyl ethyl alcohol, eugenol and others can improve olfactory ability [52,53]. Researchers have a paradoxical opinion on dietary supplementation of zinc sulfate; some found it helpful in the recovery process [54], but others rejected the benefit of prescribing zinc sulfate to catalyze the recovery process [52,53]. Therefore, further clinical studies on COVID-19 convalescent patients using mechanical and pharmaceutical methods on the use of controversial supplements such as zinc sulphate are recommended [10,55].

Understanding molecular mechanisms of smell disturbances in olfactory and trigeminal sensations may be valuable in later investigations. Identification of lesion locations (central or peripheral) by a coronavirus and specific mechanism, and further investigations to evaluate other correlations in human experiments may be valuable to suggest appropriate pharmacological and non-pharmacological treatment and preventive methods.

## 6. Conclusions

In conclusion, our study results indicate that the post-COVID smell disturbances are associated with the pathological variations in the sensitivity levels of both the olfactory and trigeminal nerves. However, the involvement of the trigeminal nerve was more prominent. Although smell perception through the olfactory and trigeminal nerves improved with time, the ability of the participants to recognize and differentiate fragrances declined. Among four main tastes, the disturbance severity of bitter taste perception was higher among the participants. Headache was associated with dysfunction of the trigeminal nerve. We also conclude that the correlation between headache and the sensitivity of smell perception via the olfactory nerve was less compared to the correlation between headache and that of the trigeminal nerve.

## Figures and Tables

**Figure 1 jcm-11-03313-f001:**
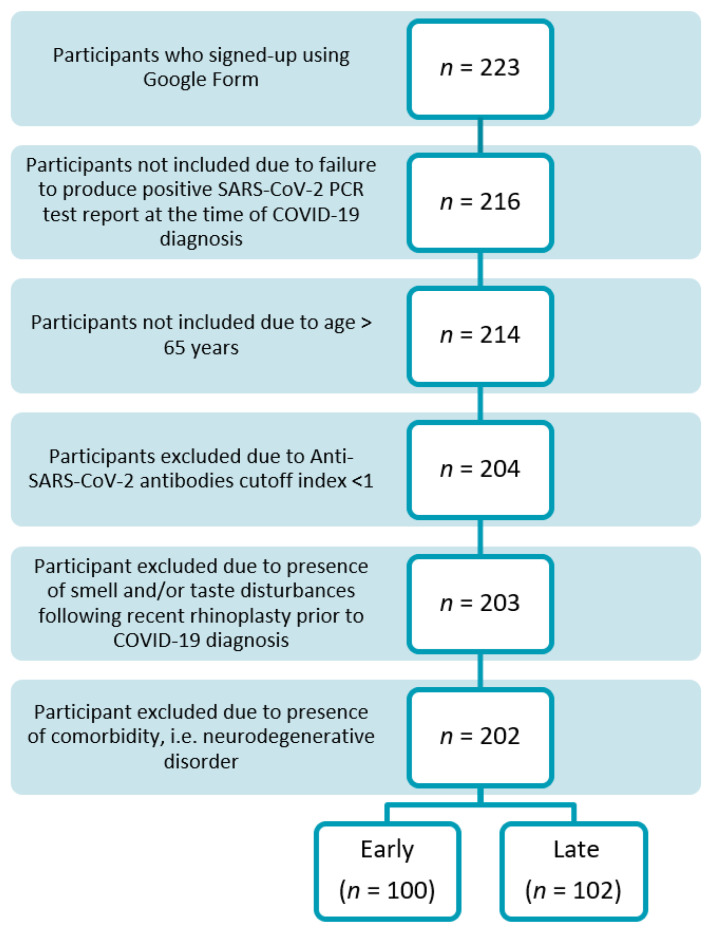
Flow chart graphically represents the study participants.

**Figure 2 jcm-11-03313-f002:**
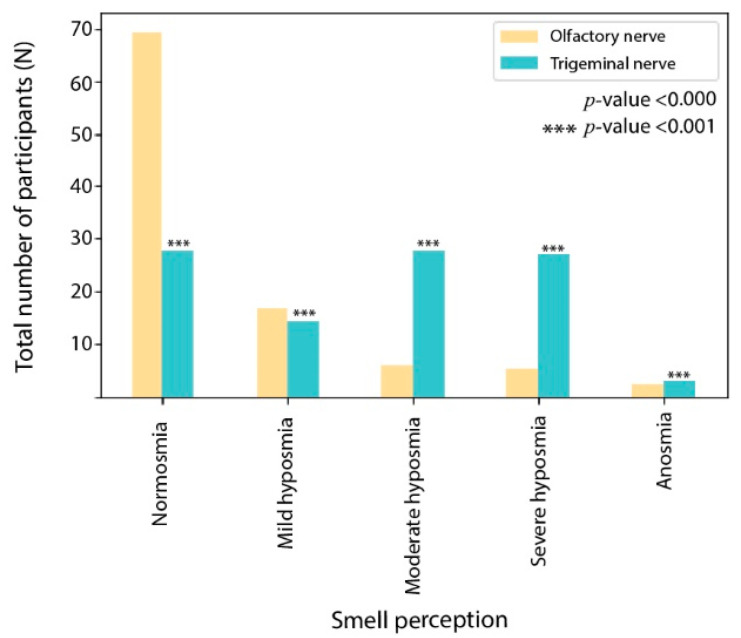
The sensory tests of the olfactory and trigeminal ability.

**Figure 3 jcm-11-03313-f003:**
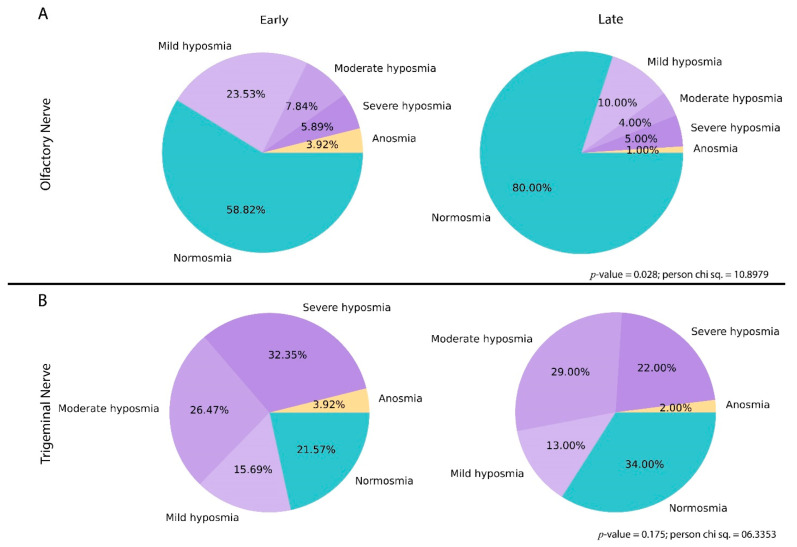
The sensory tests of the olfactory and trigeminal nerves among the participants of the early and late groups. (**A**) Results of the olfactory nerve sensory test among the early and late groups. (**B**) Results of the trigeminal nerve sensory test among the early and late groups.

**Figure 4 jcm-11-03313-f004:**
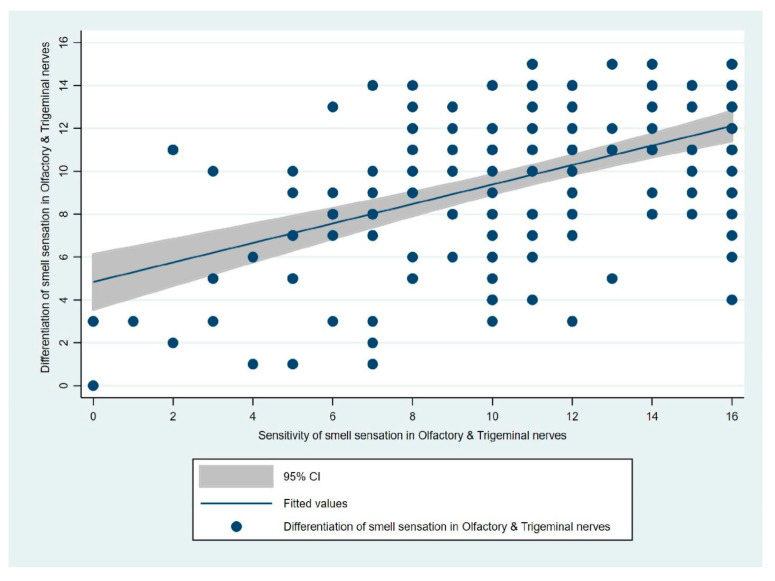
Correlation between sensitivity of smell sensation in olfactory and trigeminal nerve and differentiation capacity of smell sensation in olfactory and trigeminal nerves.

**Figure 5 jcm-11-03313-f005:**
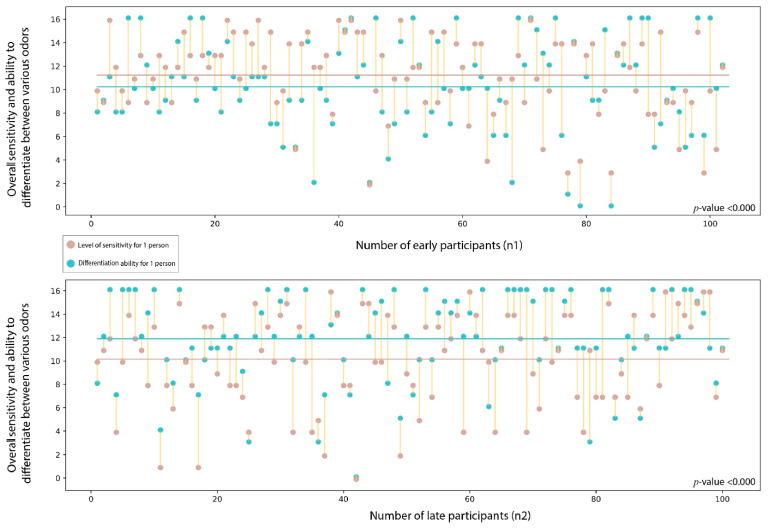
The differences in the overall sensitivity of smell and ability to differentiate between various odors as cumulatively stimulated by both the olfactory and trigeminal nerves among participants of the early and late groups.

**Figure 6 jcm-11-03313-f006:**
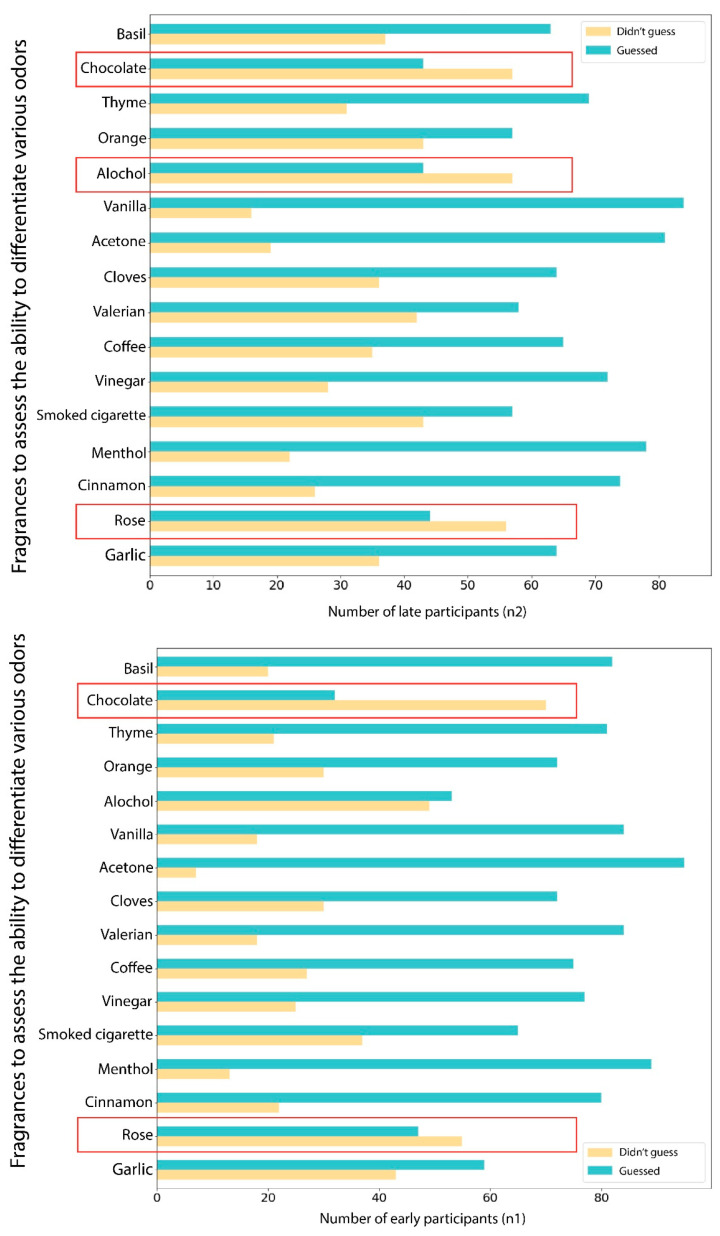
The differences in the ability to differentiate various odors among the participants of the early and late groups. The red boxes highlight fragrances with contrasting results.

**Figure 7 jcm-11-03313-f007:**
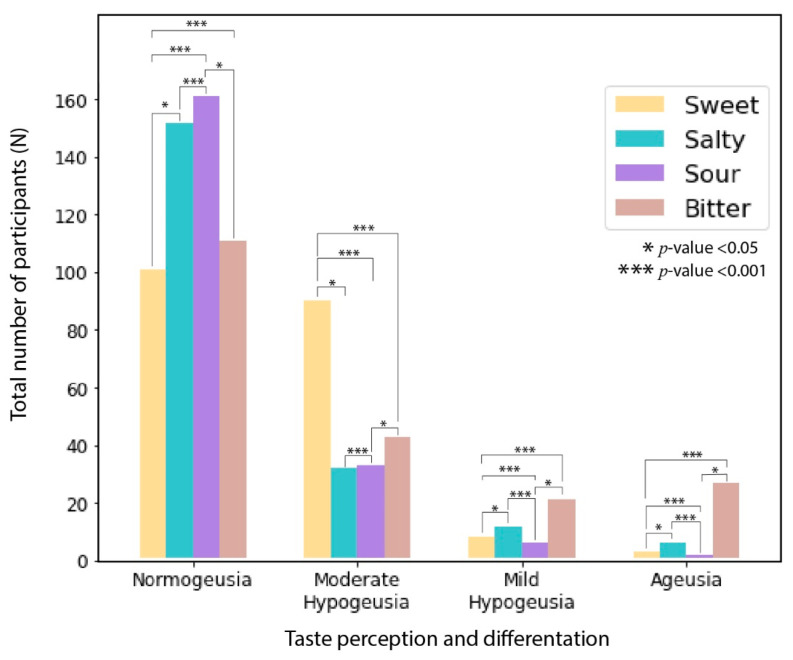
The gustatory perception and differentiation ability for different tastes.

**Figure 8 jcm-11-03313-f008:**
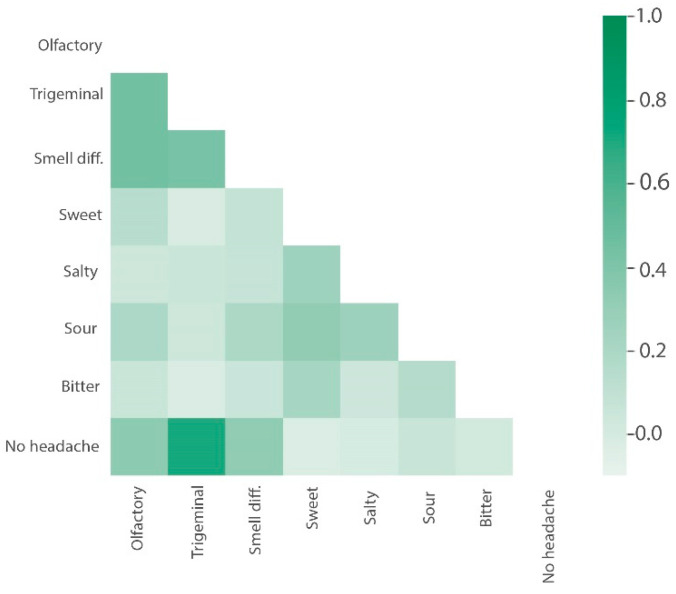
Correlations between smell sensitivity, differentiation ability, taste sensitivity and differentiation, and the prevalence of headaches.

**Table 1 jcm-11-03313-t001:** Solutions used for the assessment of taste perception and differentiation.

	Concentrated	Intermediate	Dilute
Saccharose	15 g/%	8.25 g/%	1.5 g/%
Sodium chloride	6 g/%	3.6 g/%	1.2 g/%
Citric acid	4 g/%	2.2 g/%	0.4 g/%
Caffeine benzoate	5 g/%	3.7 g/%	2.5 g/%

**Table 2 jcm-11-03313-t002:** Detailed representation of demographic data.

Descriptive Characteristics	Total Participants (*n* = 202)	Descriptive Characteristics	Total Participants (*n* = 202)
**Age in years**		**Trigeminal nerve smell status, *n* (%)**	
Mean (SD)	37.04 (11.82)	Anosmia	6 (02.97)
Min–max	18–65	Severe hyposmia	55 (27.23)
**Height in cms**		Moderate hyposmia	56 (27.72)
Mean (SD)	165.78 (07.27)	Mild hyposmia	27 (13.37)
Min–max	150–190	Normosmia	58 (28.71)
**Weight in kgs**		**Sweet taste status, *n* (%)**	
Mean (SD)	68.22 (15.47)	Ageusia/Severe hypogeusia	3 (01.49)
Min–max	41–134	Moderate hypogeusia	8 (03.96)
**BMI**		Mild hypogeusia	90 (44.55)
Mean (SD)	24.75 (5.11)	Normogeusia	101 (50.00)
Min–max	16.13–50.43	**Salty taste status, *n* (%)**	
**Sex, *n* (%)**		Ageusia/Severe hypogeusia	6 (02.97)
Male	51 (25.25)	Moderate hypogeusia	12 (05.94)
Female	151 (74.75)	Mild hypogeusia	32 (15.84)
**Olfactory nerve smell status, *n* (%)**		Normogeusia	152 (75.25)
Anosmia	5 (02.48)	**Sour taste status, *n* (%)**	
Severe hyposmia	11 (05.45)	Ageusia/Severe hypogeusia	2 (00.99)
Moderate hyposmia	12 (05.94)	Moderate hypogeusia	6 (02.97)
Mild hyposmia	34 (16.83)	Mild hypogeusia	33 (16.34)
Normosmia	140 (69.31)	Normogeusia	161 (79.70)
**Date difference (Visit—Onset of symptoms)**		**Bitter taste status, *n* (%)**	
Early (<130 days), *n* (%)	100 (49.50)	Ageusia/Severe hypogeusia	27 (13.37)
Late (>130 days), *n* (%)	102 (50.50)	Moderate hypogeusia	21 (10.40)
**Severity of disease, *n* (%)**		Mild hypogeusia	43 (21.29)
Mild	182 (90.1)	Normogeusia	111 (54.95)
Moderate	11 (5.4)		
Severe	9 (4.5)		

## Data Availability

Data can be made available by the corresponding author upon reasonable request.

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
