# Peer review of "Assessment of Smell and Taste Disturbances among COVID-19 Convalescent Patients: A Cross-Sectional Study in Armenia"

_jcm, 2022, doi:10.3390/jcm11123313_

Round 1

Reviewer 1 Report

Dear Editor,

This is a well designed clinical study that presents data about smell and taste disturbances among COVID-19 convalescent patients. The topic of the study is very interesting and add valuable contribution to the literature. Some minor points require revision,

  1. The legends of the figures are very long and includes repeated information of the text, these may be simplified
  2.  The the COVID-19 and SARS-CoV-2 abbreviations need to be write in open format first 

Author Response

Dear Reviewer,

Let us once more appreciate serious attitude to our article, which directed us to introduce the necessary changes throughout the whole paper. 

Taking into account critical observations, we made the revision of the whole sections of manuscript.

Sincerely yours,

Konstantin Yenkoyan

Here, you can find all answers to the comments considering all issues mentioned in the reviewers' comments and outline every change made point by point. The manuscript with all changes is submitted.

This is a well-designed clinical study that presents data about smell and taste disturbances among COVID-19 convalescent patients. The topic of the study is very interesting and add valuable contribution to the literature. Some minor points require revision.

1. The legends of the figures are very long and includes repeated information of the text, these may be simplified

Response: We accept the comment. The legends of figures 1, 2, 4, 5, 6 are edited. Please, find them in the revised manuscript.

2. The COVID-19 and SARS-CoV-2 abbreviations need to be written in open format first 

Response: We accept the comment.  Open format of COVID-19 and SARS-CoV-2 abbreviations have been added.

Reviewer 2 Report

This paper is about an interesting topic during the COVID-19 pandemic smell and taste disturbances. It presents though serious shortcomings in the research conducted and the presentation thereof in the manuscript.  The paper could be reconsidered in the concept of an article presenting descriptive statistics in two groups of smell and taste disturbances (early and late). The test methods used are unvalidated. According to a new submission approach, the introduction should be short and relevant and the discussion focused on the findings and not assumptive. The references MUST be updated - this is a very hot topic.

More specific comments

Abstract:

page 1 line 20 "olfactory nerve" change to "olfactory ability" or "smell"

page 1 line 20  "in the early group compared to the late group" change to  "between the early and the late group compared"

page 1 line 26 "moderately high correlation of -0.71" to be corrected  

page 1 line 28 "Pathology of the olfactory and trigeminal nerves caused disturbances in smell sensation, with the trigeminal nerve being more affected". The use of the word nerve does not correspond to the pathophysiology of the post-covid olfactory loss

Introduction

The first two paragraphs must be ommitted - two and a half years in the pandemic this is not any longer informative introduction.

page 2 line 49 "COVID-19 is also hypothesized" - change to "COVID-19 has been recognized"

page 2 line  55 "psychological impact of loss and smell and taste" change to  "psychological impact of loss of smell and taste" 

page 2 line  56-64

The WHO recognized anosmia as a COVID-19 symptom.

The lines 56-64 must be removed. We have long moved on from that point of knowledge.

page 2 line  68 "A recent study conducted..." A study published in October 2020 is not recent when we refer to the COVID-19 pandemic. 

page 2 lines 65-96 Published papers and scientific concesus on the pathophysiology of smell and taste disturbunces and on nervous system symptoms are being ignored. 

page 2 line 97 "As seen in literature" to be deleted

page 2 lines 97-112 the information is redundant - there is no need to be part of the introduction. 

The introduction needs to be reduced to the relevant to the paper 

Methodology 

page 3 lines 148. References 8-9-11 do not look relevant/right. There is an internationally accepted COVID-19 disease severity scale. The authors should use it and reference it.

page 3 lines 149. It is misleading to state that the testing was based on the UPSIT test. The authors obviously used an in-house created test. The methods sections has to present clearly how psychophysical testing was performed.

"Assessment of the ability to differentiate various odors" This test  is meant to evaluate what is widely reported as odor identification. This is not only an in-house test but it also does not have any standardization/validation reported and does not follow the rules applied by other odor identification tests 

page 5 line239  "Participants with the following characteristics were excluded from the study:  1. Anti-SARS-CoV-2 antibodies <1 cutoff index" Why were they excluded?

202 participants   who were eligible to participate were enrolled - 223 does not mean something.

Author Response

Dear Reviewer,

Let us once more appreciate serious attitude to our article, which directed us to introduce the necessary changes throughout the whole paper. 

Taking into account critical observations, we made the revision of the whole sections of manuscript.

Sincerely yours,

Konstantin Yenkoyan

Here, you can find all answers to the comments considering all issues mentioned in the reviewers' comments and outline every change made point by point. The manuscript with all changes is submitted.

Comments and Suggestions for Authors

This paper is about an interesting topic during the COVID-19 pandemic smell and taste disturbances. It presents though serious shortcomings in the research conducted and the presentation thereof in the manuscript.  The paper could be reconsidered in the concept of an article presenting descriptive statistics in two groups of smell and taste disturbances (early and late). The test methods used are unvalidated. According to a new submission approach, the introduction should be short and relevant and the discussion focused on the findings and not assumptive. The references MUST be updated - this is a very hot topic.

Response: The “Introduction” section was reduced to what we think is relevant with the context of our article. We have also refreshed the list of references with those that are most modern and correspond to the mentioned information. As for the methodology, we have responded to the comments and suggestions and made changes in the revised manuscript.

Abstract:

page 1 line 20 "olfactory nerve" change to "olfactory ability" or "smell"

Response: Comment accepted. We changed “olfactory nerve” mentioned in page 1 line 20 to “olfactory ability”

page 1 line 20  "in the early group compared to the late group" change to  "between the early and the late group compared"

Response: Thank you for the suggestions. Mentioned paragraph was re-written.

page 1 line 26 "moderately high correlation of -0.71" to be corrected  

Response: Comment accepted. We removed the word “moderately” from page 1 line 26 as well as page 13 line 420.

page 1 line 28 "Pathology of the olfactory and trigeminal nerves caused disturbances in smell sensation, with the trigeminal nerve being more affected". The use of the word nerve does not correspond to the pathophysiology of the post-covid olfactory loss

Response: Comment accepted. We have rephrased the ambiguous sentence to provide a clearer explanation.

Introduction

The first two paragraphs must be ommitted - two and a half years in the pandemic this is not any longer informative introduction.

Response: Comment accepted. The mentioned parts have been removed.

page 2 line 49 "COVID-19 is also hypothesized" - change to "COVID-19 has been recognized"

Response: Comment accepted.COVID-19 is also hypothesized" in page 2 line 49 has been replaced by “COVID-19 has been recognized”

page 2 line  55 "psychological impact of loss and smell and taste" change to  "psychological impact of loss of smell and taste" 

Response: We changed psychological impact of loss and smell and taste” in page 2 line 55 to the suggested “psychological impact of loss of smell and taste”

page 2 line  56-64

The WHO recognized anosmia as a COVID-19 symptom.

Response: We accept the related comment. page 2 line 56-64 are removed in the reviewed manuscript.

The lines 56-64 must be removed. We have long moved on from that point of knowledge.

Response: Our response to the previous point also resolves this comment.

page 2 line  68 "A recent study conducted..." A study published in October 2020 is not recent when we refer to the COVID-19 pandemic. 

Response: The word “recent” in page 2 line 68 has been removed

page 2 lines 65-96 Published papers and scientific concesus on the pathophysiology of smell and taste disturbunces and on nervous system symptoms are being ignored. 

Response: Thank you for the comment. We believe that some of the mentioned information is crucial for understanding viral infection and give a base to our study. We have reduced the text to the important points. Please find them in the revised manuscript.

page 2 line 97 "As seen in literature" to be deleted

Response: Comment accepted. We removed “as seen in literature” from page 2 line 97

page 2 lines 97-112 the information is redundant - there is no need to be part of the introduction.

Response:  page 2 lines 97-112 were reduced to an on-point explanation.

The introduction needs to be reduced to the relevant to the paper 

Response: We appreciate the comments, and we have reduced it to a shorter, more informative and topic related text. Please find it in the revised manuscript.

Methodology 

page 3 lines 148. References 8-9-11 do not look relevant/right. There is an internationally accepted COVID-19 disease severity scale. The authors should use it and reference it.

Response: We have completed the list of the mentioned references with the accepted and used WHO criteria. Please, see revised Manuscript.

page 3 lines 149. It is misleading to state that the testing was based on the UPSIT test. The authors obviously used an in-house created test. The methods sections have to present clearly how psychophysical testing was performed.

Response: We appreciate the comment on the methodology. We emphasize on the point that our method is not directly based on internationally standardized “UPSIT” test, but adapted from the methods used in them. We have chosen odors that are structurally similar to those in the before-mentioned tests, which were based on the smell perception abilities in Armenia. Because of time limitations, our questionnaire was pretested on Armenian population and the sensitivity levels for different smell and taste criteria were categorized according to the results of pretesting, authors knowledge, and standardized criteria. We added relevant information in “Methodology” section on the revised Manuscript for better understanding.

"Assessment of the ability to differentiate various odors" This test is meant to evaluate what is widely reported as odor identification. This is not only an in-house test but it also does not have any standardization/validation reported and does not follow the rules applied by other odor identification tests 

Response: Similar to other odor identification tests, the studied individuals were invited into a well-ventilated room and sat against an open window while making sure no secondary odors were interfering. We have kept 45 seconds between introducing each odor in order to eliminate the lasting effects of the previous one. The later information was also mentioned in our paper, and we can clarify more and add details if required. 

page 5 line 239  "Participants with the following characteristics were excluded from the study:  1. Anti-SARS-CoV-2 antibodies <1 cutoff index" Why were they excluded?

Response: Thank you for the comment. We will add the proper explanation next to the mentioned point on page 5 line 239. Since whenever the cutoff index is higher than 1, we can be sure that the patient was infected by SARS-CoV-2, A lesser index score means the patient had no antibodies, but it does not mean that they weren’t infected. So, it was a reassurance that the cause of the olfactory and gustatory disturbances was surely COVID-19. 

202 participants   who were eligible to participate were enrolled - 223 does not mean something.

Response: The participants count was updated in order to keep only the enrolled ones.

Reviewer 3 Report

Dear Author/Editor,

Thank for conducting this study related to covid-19 severity/intensity of disease with the neurological evaluation of patients. Here the authors assessed the degree of smell and taste disturbances among Armenian Covid-19 patients. The authors claimed the study design as cross-sectional, however, I see it was case-control study according to study setup and patients’ recruitment. Are all the patients for both groups recruited randomly then divided into two categories based on their course of disease or they had been enrolled based on their disease states for two different groups??? If the second one is true for this study, I think it would be better to mention as case-control study design. Please clarify this issue. Also, I have some other observations,

  1. The authors presented their study results in a descriptive way, is it possible to find any correlations/confounding factors for those neurological problems?
  2. The authors are asked to propose some recommendations to avoid/decrease the intensity of neurological problems among Covid-19 patients.
  3. How these problems can be treated, is there any information regarding this issues.

Author Response

Dear Reviewer,

Let us once more appreciate serious attitude to our article, which directed us to introduce the necessary changes throughout the whole paper. 

Taking into account critical observations, we made the revision of the whole sections of manuscript.

Sincerely yours,

Konstantin Yenkoyan

Here, you can find all answers to the comments considering all issues mentioned in the reviewers' comments and outline every change made point by point. The manuscript with all changes is submitted.

Comments and Suggestions for Authors

Thank for conducting this study related to covid-19 severity/intensity of disease with the neurological evaluation of patients. Here the authors assessed the degree of smell and taste disturbances among Armenian Covid-19 patients. The authors claimed the study design as cross-sectional, however, I see it was case-control study according to study setup and patients’ recruitment. Are all the patients for both groups recruited randomly then divided into two categories based on their course of disease or they had been enrolled based on their disease states for two different groups??? If the second one is true for this study, I think it would be better to mention as case-control study design. Please clarify this issue. Also, I have some other observations,

Response: Thank you for your comment. We would like to clarify that we recruited all the participants randomly and then categorized them into two groups based on their course of the disease. Hence, we consider our research design to be a cross-sectional study and not a case-control study.

1. The authors presented their study results in a descriptive way, is it possible to find any correlations/confounding factors for those neurological problems?

Response: Indeed, there can be correlation and confounding between various neurological problems, but since we only collected subjective opinions, we were not able to derive reliable results in this paper. Regardless, headache was one of the neurological problems which was included in our patient questionnaire acquired by the researchers, and a visible high correlation was reported in our results.

2. The authors are asked to propose some recommendations to avoid/decrease the intensity of neurological problems among Covid-19 patients.

Response: We appreciate your comment. A recommendation section is added to the revised manuscript mentioning further studies for neurological problems. Due to lacking data, as well as our efforts to find sufficient information on how to deal with COVID associated neurological problem, we will give recommendations according to available literature.

3. How these problems can be treated, is there any information regarding this issues.

Response: The recommendations section contains information regarding this. Please, find it in the revised Manuscript.

This manuscript is a resubmission of an earlier submission. The following is a list of the peer review reports and author responses from that submission.